# Differentially Represented Proteins in Response to Infection with *Mycobacterium tuberculosis* Identified by Quantitative Serum Proteomics in Asian Elephants

**DOI:** 10.3390/pathogens11091010

**Published:** 2022-09-03

**Authors:** Margarita Villar, Rajesh Man Rajbhandari, Sara Artigas-Jerónimo, Marinela Contreras, Amir Sadaula, Dibesh Karmacharya, Paulo Célio Alves, Christian Gortázar, José de la Fuente

**Affiliations:** 1SaBio, Instituto de Investigación en Recursos Cinegéticos IREC-CSIC-UCLM-JCCM, Ronda de Toledo s/n, 13005 Ciudad Real, Spain; 2Biochemistry Section, Faculty of Science and Chemical Technologies, and Regional Centre for Biomedical Research (CRIB), University of Castilla-La Mancha, 13071 Ciudad Real, Spain; 3Center for Molecular Dynamics Nepal (CMDN), Thapathali Road 11, Kathmandu 44600, Nepal; 4Dep. de Biologia, Faculdade de Ciencias da Universidade do Porto, Rua do Campo Alegre, S/N, Edificio FC4, 4169-007 Porto, Portugal; 5CIBIO, Centro de Investigação em Biodiversidade e Recursos Genéticos, InBIO Laboratório Associado—BIOPOLIS Program in Genomics, Biodiversity and Land Planning, Campus de Vairão, Universidade do Porto, 4485-661 Vairão, Portugal; 6National Trust for Nature Conservation, Biodiversity Conservation Center, Sauraha, Chitwan 44204, Nepal; 7Estação Biológica de Mértola EBM, Praça Luís de Camões, Mértola, 7750-329 Mértola, Portugal; 8Department of Veterinary Pathobiology, Center for Veterinary Health Sciences, Oklahoma State University, Stillwater, OK 74078, USA

**Keywords:** elephant, tuberculosis, proteomics, biomarkers, vaccine, diagnostic

## Abstract

Tuberculosis is a major global concern. Tuberculosis in wildlife is a risk for zoonotic transmission and becoming one of the challenges for conservation globally. In elephants, the number of cases is likely rising. The aim of this study was to identify proteins related to tuberculosis infection in elephants, which could then be used for the development of diagnostic tools and/or vaccines. A serum proteomics approach was used to characterize differentially represented proteins in response to *Mycobacterium tuberculosis* in Asian elephants (*Elaphas maximus*). Blood samples were collected from eight elephants, four of which were antibody positive for tuberculosis and four were antibody negative. Proteomics analysis identified 26 significantly dysregulated proteins in response to tuberculosis. Of these, 10 (38%) were identified as immunoglobulin and 16 (62%) as non-immunoglobulin proteins. The results provided new information on the antibody response to mycobacterial infection and biomarkers associated with tuberculosis and protective response to mycobacteria in Asian elephants. Protective mechanisms included defense against infection (Alpha-1-B glycoprotein A1BG, Serpin family A member 1 SERPINA1, Transthyretin TTR), neuroprotection (TTR), and reduced risks of inflammation, infections, and cancer (SERPINA1, Keratin 10 KRT10). Using a translational biotechnology approach, the results provided information for the identification of candidate diagnostic, prognostic, and protective antigens for monitoring and control of tuberculosis in Asian elephants.

## 1. Introduction

Tuberculosis (TB), a disease caused by organisms belonging to the *Mycobacterium tuberculosis* complex (MTBC), remains a major global concern for public health and the economy. With an estimated 10 million cases worldwide and 1.5 million deaths in humans in the year 2020, it is the second infectious disease with global impact after COVID-19 [1]. Tuberculosis in humans and animals is caused by a group of closely related acid-fast bacilli of the MTBC [2,3]. The MTBC includes *M. tuberculosis, M. bovis, M, bovis* (bacillus Calmette- Guerin)*, M. africanum, M, caprae, M. microti, M. canettii and M. pinnipedii* [4]. In elephants, TB was probably present more than 2000 years ago [5] with the first case of TB in elephants reported in the London Zoo in 1875 [6]. The main causative agent for TB in elephants is *M. tuberculosis* [7] with rare cases of other MBTC organisms such as *M. bovis* [8]. The sporadic cases of TB in captive Asian elephants (*Elaphas maximus*) were reported in the early 20th century but only in the mid-20th century the first case in an African elephant (*Loxodonta africana*) was reported [8]. Since the initiation of systematic surveillance in 1998 in the U.S.A., the number of TB cases in elephants is rising [8].

Nepal is home to Asian elephants that are still used in the timber industry for military and national park security and for tourism [9,10,11,12]. At present, there are altogether 208 captive elephants in Nepal out of which 94 government-owned elephants are in various parks in the Terai, namely Koshi Tappu Wildlife Reserve, Parsa Wildlife Reserve, Chitwan National Park, Bardia National Park, and Suklaphanta Wildlife Reserve [9]. In Nepal, TB caused by different members of the MTBC has been reported in different animal species including *M. tuberculosis* in Asian elephants [13,14,15].

Tuberculosis in wildlife is becoming one of the challenges for conservation globally. The disease spillover and transmission on the human–livestock–wildlife interface affects a wide range of species [16]. Close contact between elephants and humans, especially between captive elephants and their mahouts, increases the risk of disease transmission. Furthermore, studies that have been carried out in Nepal have shown that there might be an interspecies transmission of TB among human and animal species with a recent study on bovine TB revealing a high risk of zoonotic bovine TB at the cattle–human interface [17,18].

Regarding the diagnosis of TB in elephants, a wide variety of diagnostic measures have been used throughout the years. Culture of the trunk wash sample was preferred as a diagnostic tool before more rapid methods became available. Elephant TB can be accurately diagnosed by serum antibody detection [19,20,21,22]. In the absence of highly sensitive diagnostic assays, a routine medical examination is recommended [23]. Standard tests such as VetMAX MTBC qPCR Kit [24] and serological tests using Elephant TB STAT-PAK, DPP VetTB Assay, MAPIA (multi-antigen print immunoassay) [22], and interferon gamma release assay (IGRA) [25] are used for detection of MTBC in different parts of the world depending on sample accessibility.

The main challenge to further advances in TB diagnosis and control is the identification of biomolecules for developing diagnostic tools and vaccines. Few studies have used proteomics approaches for the characterization of protein profiles in elephantids [26,27,28]. However, the study reported here is the first characterization of elephant serum proteome in response to TB. These results advance our understanding of elephant immune response to *M. tuberculosis* and provide potential biomarkers for diagnosis and control of TB in this endangered species.

## 2. Materials and Methods

### 2.1. Serum Samples from TB-Positive (TB+) and TB-Negative (TB−) Elephants

The study was conducted in Chitwan National Park established in 1973 in Nepal. It covers an area of 952.63 km^2^ and is in the subtropical Inner Terai lowlands of south-central Nepal in the districts of Nawalpur, Parsa, Chitwan, and Makwanpur. In altitude, it ranges from about 100 m in the river valleys to 815 m in the Churia Hills (Shrestha and Shrestha, 2021). The National Park is home to 111 captive elephants. Of them, serum samples were collected from 8 adult animals with (E1, E4, E44, E66; N = 4) and without (E9, E43, E70, E71; N = 4) tuberculosis reactive antibodies and used for the proteomics analysis. Blood samples were collected by trained veterinarians from an ear vein into a serum gel tube, using the Vacutainer(R) blood collection system with butterfly needle. Serum samples were aliquoted and stored at −20 °C until analysis. Ethical approval for the study was obtained from Nepal Ethical Review Board, Nepal Health Research Council (IRC number 312/2018).

### 2.2. Diagnostic of Tuberculosis

Sera were subjected to antibody testing using an animal-side rapid test for TB (DPP VetTB Assay, Chembio Diagnostic Systems Inc., Medford, MA, USA) performed as recommended by the manufacturer. Results showed the presence of TB reactive or non-reactive antibodies. Two samples from TB+ animals E1 and E44 were further confirmed as infected with *M. tuberculosis* by cell culture.

### 2.3. Serum Proteomics

The serum proteomics methodology has been previously reported [29] and adapted to this study. Protein concentration in serum samples was determined using the BCA protein assay with BSA (Sigma-Aldrich, St. Louis, MO, USA) as standard. Protein serum samples (150 µg per sample) were trypsin digested using the FASP Protein Digestion Kit (Expedeon Ltd., UK) and sequencing grade trypsin (Promega, Madison, WI, USA) following manufacturer´s recommendations. The resulting tryptic peptides were desalted onto OMIX Pipette tips C18 (Agilent Technologies, Santa Clara, CA, USA), dried down, and stored at −20 °C until mass spectrometry analysis. The desalted protein digests were resuspended in 2% acetonitrile, and 5% acetic acid in water and analyzed by reverse phase liquid chromatography coupled to mass spectrometry (RP-LC-MS/MS) using an EkspertTM nanoLC 415 system coupled online with a 6600 TripleTOF® mass spectrometer (AB SCIEX; Framingham, US) through information-dependent acquisition (IDA) followed by sequential windowed data independent acquisition of the total high-resolution mass spectra (SWATH-MS). The peptides were concentrated in a 0.1 × 20 mm C18 RP precolumn (Thermo Scientific) with a flow rate of 2µl/min for 10 min in solvent A. Then, peptides were separated in a 0.075 × 250 mm C18 RP column (New Objective, Woburn, MA, USA) with a flow rate of 300 nl/min. Elution was performed in a 120 min gradient from 5% B to 30% B followed by 15 min gradient from 30% B to 60% B (Solvent A: 0.1% formic acid in water, solvent B: 0.1% formic acid in acetonitrile) and directly injected into the mass spectrometer for analysis. For IDA experiments, the mass spectrometer was set to scan full spectra from 350 *m*/*z* to 1400 *m*/*z* (250 ms accumulation time) followed by up to 50 MS/MS scans (100–1500 *m*/*z*). Candidate ions with a charge state between +2 and +5 and counts per second above a minimum threshold of 100 were isolated for fragmentation. One MS/MS spectrum was collected for 100 ms, before adding those precursor ions to the exclusion list for 15 s (mass spectrometer operated by Analyst® TF 1.6, ABSciex®). Dynamic background subtraction was turned off. Data were acquired in high sensitivity mode with rolling collision energy on and a collision energy spread of 5. An equal amount of the four samples for each control and infected group joined together as a representative mixed sample, which was used for the generation of the reference spectral ion library as part of SWATH-MS analysis. A total amount of 4 µg was injected by triplicate. For SWATH quantitative analysis, 8 independent samples (2 technical replicates from each of the 4 biological replicates for TB− and TB+ groups) (6 μg each) were subjected to the cyclic data-independent acquisition (DIA) of mass spectra using the SWATH variable windows calculator (V 1.0, AB SCIEX) and the SWATH acquisition method editor (AB SCIEX), like established methods [28]. A set of 50 overlapping windows was constructed (containing 1 *m*/*z* for the window overlap), covering the precursor mass range of 400–1250 *m*/*z*. For these experiments, a 50 ms survey scan (350–1400 *m*/*z*) was acquired at the beginning of each cycle, and SWATH-MS/MS spectra were collected from 100–1500 *m*/*z* for 70 ms at high sensitivity mode, resulting in a cycle time of 3.6 s. Collision energy for each window was determined according to the calculation for a charge +2 ion-centered upon the window with a collision energy spread of 15. To create a spectral library of all the detectable peptides in the samples, the IDA MS raw files were combined and subjected to database searches in unison using ProteinPilot software v. 5.0.1 (AB SCIEX) with the Paragon algorithm. Spectra identification was performed by searching against the Uniprot Elephantidae database (28,137 entries in April 2022) with the following parameters: iodoacetamide cysteine alkylation, trypsin digestion, identification focus on biological modification, and thorough ID as search effort. The detected protein threshold was set at 0.05. To assess the quality of identifications, an independent false discovery rate (FDR) analysis with the target-decoy approach provided by Protein PilotTM was performed. Positive identifications were considered when identified proteins reached a 1% global FDR. The mass spectrometry proteomics data have been deposited in the ProteomeXchange Consortium via the PRIDE partner repository with the dataset identifier PXD033830 and 10.6019/PXD033830.

### 2.4. Quality Control of Proteomics Data

As previously validated [29], quality of proteomics data was controlled at multiple levels. First, a Zebrafish intestine digest was used for the evaluation of instrument performance. Buffer A samples were run as blanks every two injections to prevent carry-over. Two technical replicates were injected for each sample.

### 2.5. Data Analysis

As previously reported for SWATH processing [29], up to 10 peptides with 7 transitions per protein were automatically selected by the SWATH Acquisition MicroApp 2.0 in the PeakView 2.2 software with the following parameters: 15 ppm ion library tolerance, 5 min XIC extraction window, 0.01 Da XIC width, and considering only peptides with at least 99% confidence and excluding those which were shared or contained modifications. However, to ensure reliable quantitation, only proteins with 3 or more peptides available for quantitation were selected for XIC peak area extraction and exported for analysis in the MarkerView 1.3 software (AB SCIEX). Global normalization according to the total area sums of all detected proteins in the samples was conducted (Appendix A). The Student’s *t*-test (*p* < 0.05) was used to perform two-sample comparisons between the averaged area sums of all the transitions derived for each protein across the eight replicate runs for each group under comparison, to identify proteins that were significantly differentially represented between TB− and TB+ elephants (Appendix A). Protein log fold change relative intensity was compared between TB− and TB+ elephants by Welch’s unpaired *t*-test (*p* < 0.05; https://www.graphpad.com/quickcalcs/ttest1/?Format=C) [29,30]. Data were separately analyzed for overrepresented and underrepresented proteins. For differentially represented immunoglobulin (Ig) proteins (Igs), an analytical algorithm was developed using Protein BLAST sequence alignment against non-redundant protein database(nr) using compositional matrix adjustment (https://blast.ncbi.nlm.nih.gov/Blast.cgi?PROGRAM=blastp&PAGE_TYPE=BlastSearch&LINK_LOC=blasthome) and Paratome (http://www.ofranlab.org; [31]) and used for the identification of *Mycobacterium* antigen binding regions and candidate biomarkers of protective response to mycobacteria (Appendix A). Phylogenetic analysis was conducted for *M. tuberculosis* type I site-specific deoxyribonuclease, HsdR family (sequence ID: SGD45129.1, length: 1040 amino acids) using BLAST Tree Viewer, Fast Minimum Evolution, Max Seq Difference = 0.85, Distance Grishin Protein. For non-Ig proteins, gene ontology (GO) annotations for molecular function (MF), biological process (BP), and subcellular localization (CC) were performed in human homolog proteins using Uniprot (https://www.uniprot.org), and the most significant pathways were identified by Reactome (https://reactome.org/PathwayBrowser/; [32,33]) (Appendix A). COBALT (https://www.ncbi.nlm.nih.gov/tools/cobalt/cobalt.cgi?CMD=Web) was used for protein sequence alignment between identified mycobacterial proteins with predicted antigen binding regions to Ig proteins and *M. tuberculosis* P9WNK5|ESXB_MYCTU ESAT-6-like protein EsxB identified as highly reactive to antibodies in infected elephants [34].

## 3. Results

### 3.1. Identification of Differentially Represented Elephant Serum Proteins in Response to TB by Quantitative Serum Proteomics

In our study, sera were collected from eight Asian elephants, four of which were seronegative for TB, and four of which were seropositive for TB, and used individually for SWATH-MS proteomics analysis for the identification and quantitation of differentially represented proteins (Figure 1A). Proteomics analysis identified 26 significantly dysregulated proteins in TB+ elephants when compared to TB− animals (Figure 1B, Appendix A). Of them, 10 (38%) were identified as Ig proteins (nine overrepresented and one underrepresented in TB+ elephants) and sixteen (62%) as non-Ig proteins (twelve overrepresented and four underrepresented in TB+ elephants) (Figure 1B). Data analysis was then focused on overrepresented and underrepresented Ig and non-Ig proteins for the identification of candidate protective or disease-associated biomarkers (Figure 1A–C).

### 3.2. Characterization of Ig Protein Profiles and Correlation with Mycobacterial Infection or Host Protective Capacity

Changes in Ig protein mass spectra relative intensity in individual samples were compared between TB− and TB+ elephants. The results were significantly different between groups for both overrepresented and underrepresented proteins thus further supporting proteomics results (*p* < 0.05; Figure 2).

An analytical workflow was developed to characterize the Ig-like domain-containing proteins identified as significantly dysregulated in TB+ elephants when compared to TB− animals (Figure 3, Appendix A). The use of the Paratome algorithm (http://www.ofranlab.org) allows the identification of antigen binding regions in identified Ig light and heavy chain variable regions [30]. 

The predictive models for Igs overrepresented in TB+ elephants showed an association with exposure to different *Mycobacterium* spp. (Figure 3). As demonstrated by an example analysis using the identified *M. tuberculosis* type I site-specific deoxyribonuclease HsdR family protein (SGD45129.1; Appendix A), these mycobacteria are phylogenetically closely related to these proteins (Figure 3). Only one Ig protein was identified as underrepresented in TB+ elephants and thus overrepresented in TB− animals (G3UIN4; Appendix A) with predicted *Mycobacterium* sp. reactive epitopes associated with biomarkers of a protective response to mycobacteria (Figure 3).

### 3.3. Characterization of Non-Ig Protein Profiles and Correlation with TB

The analysis of non-Ig protein profiles in response to TB in elephants resulted in the identification of proteins in multiple biological processes and pathways associated with pathogenic and protective mechanisms (Figure 4; Appendix A). Most of the dysregulated proteins were secreted but some were also located in the cytoplasm, endoplasmic reticulum (ER), and cytoskeleton of multiple cell types (Figure 4; Appendix A).

Dysregulated proteins in TB+ elephants were mostly associated with higher risks of disorders/diseases such as glycogen storage disease type Ia (Biotinidase, BTD), venous thrombosis, and hypercoagulability (joining the chain of multimeric IgA and IgM, JCHAIN; immunoglobulin heavy constant mu, IGHM; Christmas factor IX, F9; glycoprotein V platelet, GP5; complement C5, C5), postprandial hypertriglyceridemia, heart disease, cardiovascular events and mortality (apolipoproteins C-III, APOC3; D, APOD; B, APOB), solid malignancies (Tenascin C, TNC), pathogenic bacteria-induced pneumonia (CD5 molecules such as CDL5), and inflammation and atherosclerosis (glycosyl-phosphatidylinositol-specific phospholipase D, GPLD1) (Figure 4; Appendix A). 

However, the non-Ig proteins identified as overrepresented in TB− elephants and thus underrepresented in TB+ animals were associated with possible protective mechanisms such as defense against infection and reduced risk of inflammation (Figure 4; Appendix A). 

### 3.4. Implications for Translational Biotechnology

The results presented here provide information for the identification of candidate diagnostic, prognostic, and protective antigens for TB in Asian elephants.

**Candidate diagnostic antigens.** Mycobacterial proteins reactive to Ig proteins in TB+ elephants may constitute candidate diagnostic antigens. For example, the highly represented Ig protein, G3UDI6 (TB+/TB− ratio = 1.9; Appendix A), reacted to *Mycobacterium* sp. DUF3298 domain-containing protein (TAM68061.1), FkbM family methyltransferase (WP_020822873.1) and hypothetical proteins OEM_30260 and MOTT12_03025 (AGP64561.1 and ARR78689.1) (Appendix A). Therefore, these antigens may be used for the diagnosis of TB in Asian elephants. Furthermore, the combination of identified reactive epitopes may be used for designing a diagnostic chimeric antigen (ISSANYG-GGGGS-WLRYYSDRNWNR-C-KLH) with a putative higher capacity to diagnose TB+ elephants (Figure 5). The proposed candidate diagnostic chimeric antigen protein BLAST sequence alignment against mycobacteria nr database showed 31.6 to 78.9% sequence identity to environmental mycobacteria (Figure 5). Mycobacteria with the highest sequence identity in the *M. avium* complex also contain pathogenic bacteria found in elephants [34]. These results support the possibility of using this chimeric antigen combined with other tests for the diagnosis of TB in elephants. 

**Candidate prognostic antigens.** Identified non-Ig proteins in Asian elephants may constitute TB disease prognostic antigens. For example, TNC (TB+/TB− ratio = 2.7) to KRT10 (TB+/TB− ratio = 0.8) serum protein levels (TNC/KRT10 ratio = 0.7) (Appendix A) may be used to evaluate the risks of developing solid malignancies in infected animals (Figure 5). With results for individual animals included in this study (Appendix A), the TNC/KRT10 ratio was lower in TB− (0.1–0.3) than in TB+ elephants (0.5–1.1) with animal No. 66 showing a predicted high risk for developing solid malignancies (proposed threshold at TNC/KRT10 ratio = 1) (Figure 5). 

**Candidate protective antigens.** The identified underrepresented Ig protein in TB+ elephants and thus overrepresented in TB− animals (G3UIN4; Appendix A) reacted to *Mycobacterium* sp. epitopes with predicted protective capacity (Figure 3). Therefore, the identification and combination of reactive epitopes in mycobacterial RND family transporter membrane protein and protein translocase subunit SecD (Appendix A) may be used to design a new candidate protective chimeric antigen with predicted reactive immunological quantum or protective epitopes (QSLFDATDKID-FDSS-KNDEFFYL-GGGGS-LIYLASK-C-KLH) for TB control in Asian elephants (Figure 5).

## 4. Discussion

The Igs overrepresented in TB+ elephants were associated with exposure to different *Mycobacterium* spp. including *M. tuberculosis* (Ig reactive to HsdR and DinB family protein; MBZ4317686.1), *Mycobacterium intracellulare* or *Mycobacterium avium* complex (FkbM family methyltransferase; WP_020822873.1), *Mycobacterium asiaticum* (DUF2617 family protein; WP_065128554.1) and *Mycobacterium colombiense* (hypothetical protein; WP_076051897.1) (Appendix A) that have been previously identified in elephants [35,36,37]. Other *Mycobacterium* proteins such as ESAT-6-like protein EsxB (CFP10/ESAT-6) reported as highly reactive to antibodies in infected elephants [34] were not identified as predicted antigen binding regions in our study. Nevertheless, cross-reactive epitopes between CFP10/ESAT-6 and identified mycobacterial proteins with predicted antigen binding regions to Ig proteins may be present (Appendix A). These animals are exposed to MTBC and environmental mycobacteria such as *Mycobacterium simiae* and *M. avium* complex identified here and therefore antibodies induced by bacterial cross-reactive antigens may be present [38].

For the only identified Ig protein in TB+ overrepresented in TB− animals (G3UIN4; Appendix A), predicted *Mycobacterium* sp. reactive epitopes were associated with biomarkers of a protective response to mycobacteria associated with (a) resistance-nodulation-division (RND) family transporter (WP_067937080.1) membrane protein associated with drug resistance [39,40], (b) restriction endonuclease/methylase (TSD87553.1) involved in the anti-mycobacterial activity in response to vaccination that correlates with altered DNA methylation [41], and protein translocase subunit SecD (MBV9350256.1), a transmembrane transport cell inner membrane protein involved in secretion of mycobacterial proteins linked to pathogenicity [42,43] (Appendix A).

These results provided new information on the antibody response to mycobacterial infection in Asian elephants. While TB+ elephants showed Ig proteins cross-reactive to various *Mycobacterium* spp., the results suggested that TB− elephants developed a protective antibody response against mycobacterial proteins involved in disease-associated mechanisms.

Dysregulated non-Ig proteins in TB+ elephants were mostly associated with higher risks of disorders/diseases [44,45,46,47,48]. The most represented biological processes and pathways were related to cardiovascular disorders/diseases with nine out of twelve of the overrepresented proteins in TB+ elephants (Figure 4). Cardiac disorders are typical in TB [49]. Elevated levels of vitamin K-dependent protein F9 have been associated with an increased risk of venous thrombosis [50]. The GP5-mediated adhesion of platelets to injured vascular surfaces in the arterial circulation is a critical initiating event in hemostasis and apolipoproteins (APOs) may increase the risk of postprandial hypertriglyceridemia associated with cardiovascular disease [51]. Elevated plasma concentration of complement factor C5, which stimulates the locomotion of polymorphonuclear leukocytes toward sites of inflammation is associated with the risk of venous thromboembolism [52]. Although the association between cardiovascular disorders and TB has not been documented in elephants, as in other species, these results suggest a higher risk for these disorders in TB+ Asian elephants [53,54,55,56].

Possible protective mechanisms associated with non-Ig proteins overrepresented in TB− elephants included defense against infection (Alpha-1-B glycoprotein A1BG, Serpin family A member 1 SERPINA1, Transthyretin TTR), neuroprotection (TTR), and reduced risks of inflammation, infections, and cancer (SERPINA1, Keratin 10 KRT10) [57,58,59,60,61]. 

The application of translational biotechnology approaches would advance the control of infectious diseases as described in a previous study on tick-borne diseases [62]. A candidate diagnostic chimeric antigen was designed based on mycobacterial proteins reactive to Ig proteins in TB+ elephants while candidate prognostic antigens were proposed using identified non-Ig proteins in Asian elephants (Figure 5). However, infection with other pathogens may also affect serum levels of these proteins and thus the proposed candidate diagnostic antigen requires additional experiments for validation in TB versus other common infections in elephants. Although elephants appear to show a genetic-based lower than expected incidence of cancer when compared to humans [63,64], the association of solid malignancies with TB may constitute a risk factor to be evaluated using the prognostic antigens proposed here. The application of the quantum vaccinomics approach [65,66] guided the design of a candidate protective antigen for vaccination against TB in Asian elephants (Figure 5). Based on the challenge that represents a better understanding of the protective mechanisms against TB [67], the proposed candidate protective chimeric antigen is only speculative and requires validation in vaccination studies. 

In conclusion, the results of the Asian elephant serum proteome in response to TB advanced our understanding of the elephant immune response to *M. tuberculosis*. Future studies should further explore the design, production, and essay of identified candidate diagnostic, prognostic and protective antigens for TB monitoring and control in Asian elephants.

## Figures and Tables

**Figure 1 pathogens-11-01010-f001:**
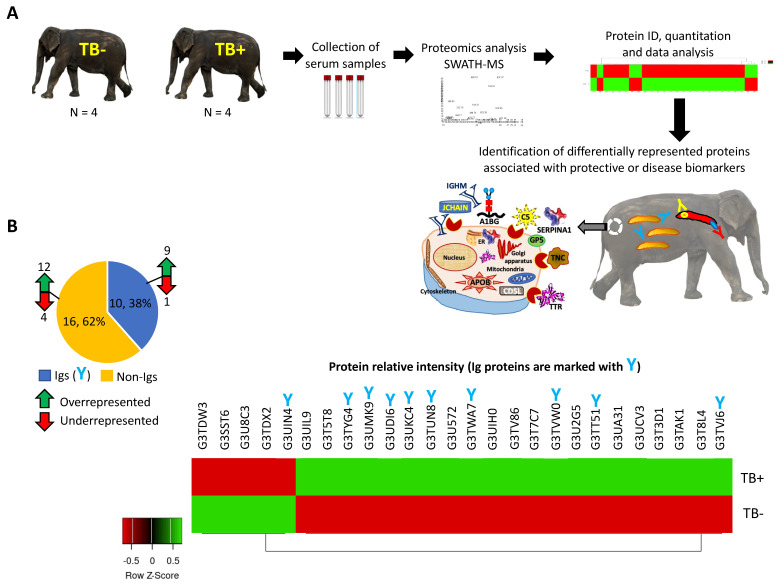
Identification of serum differentially represented proteins in response to TB in Asian elephants. (**A**) In our study, sera were collected from TB− and TB+ elephants. Sera were subjected to SWATH-MS quantitative proteomics to characterize serum differentially protein profiles. The proteomics results were then focused on overrepresented and underrepresented Ig and non-Ig proteins and translated into the identification of candidate protective or disease-associated biomarkers. (**B**) Number and heatmap (Z-scored relative intensity original value) of overrepresented and underrepresented Ig and non-Ig proteins in TB+ elephants when compared to TB− animals.

**Figure 2 pathogens-11-01010-f002:**
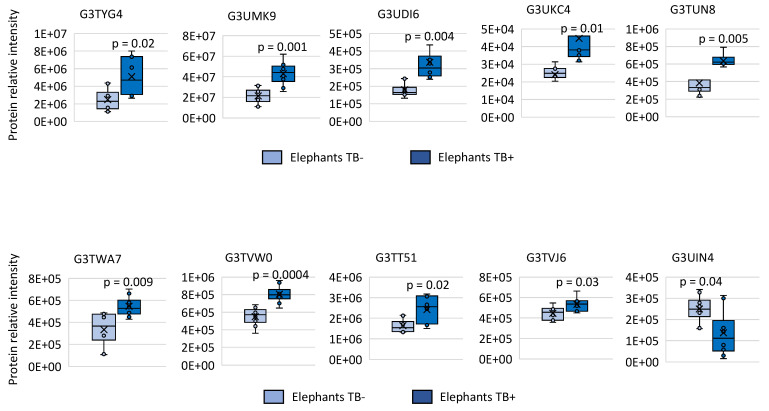
Serum Ig proteins differentially represented in TB− and TB+ elephants. Changes in Ig protein mass spectra relative intensity in individual samples of all TB− and TB+ animals. Relative intensity was compared between TB− and TB+ cohorts by Welch’s unpaired *t*-test (*p* < 0.05; *n* = 4 biological replicates).

**Figure 3 pathogens-11-01010-f003:**
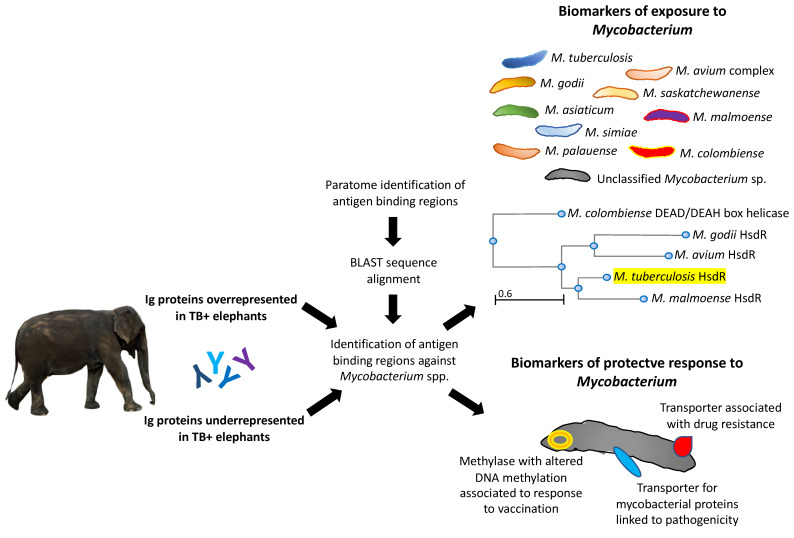
Predicted biomarkers associated with differentially represented Ig proteins. Analytical workflow based on Paratome and protein BLAST sequence alignment identified biomarkers associated with exposure to *Mycobacterium* spp. or protective response to mycobacteria in Ig proteins overrepresented and underrepresented in TB+ elephants, respectively. Phylogenetic analysis was conducted for *M. tuberculosis* type I site-specific deoxyribonuclease, HsdR family (SGD45129.1) input sequence highlighted in yellow using BLAST Tree Viewer, Fast Minimum Evolution, Max Seq Difference = 0.85, Distance Grishin Protein.

**Figure 4 pathogens-11-01010-f004:**
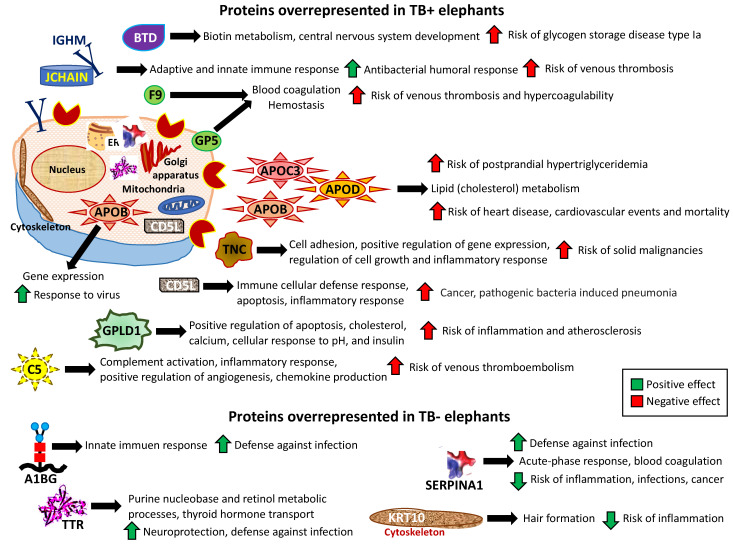
Predicted correlates with protective or disease-associated pathways in non-Ig differentially represented proteins. Analytical workflow based on Uniprot GO annotations and Reactome identified biomarkers of protective or disease-associated pathways in response to TB in elephants.

**Figure 5 pathogens-11-01010-f005:**
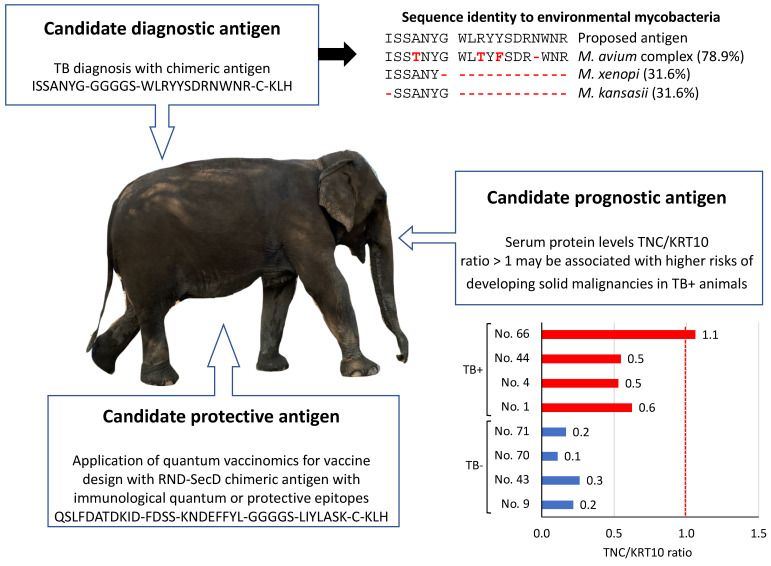
Translational biotechnology approaches applied to the differentially represented serum proteins identified in TB+ and TB− Asian elephants. For candidate diagnostic antigen, sequence alignment was conducted against environmental mycobacteria using BLAST (percent sequence identity with non-identical amino acids in red are shown). For candidate prognostic antigens, results are shown for TNC/KRT10 ratio (average of two technical replicates) in individual animals included in this study (Appendix A). Elephant No. 66 showed a predicted high risk for developing solid malignancies (red line for proposed threshold at TNC/KRT10 ratio = 1). In candidate diagnostic and protective antigens, GGGGS and C-KLH are linked peptide sequences not derived from mycobacterial proteins. The results of the study provided information for the design of candidate diagnostic, prognostic, and vaccine antigens.

## Data Availability

The mass spectrometry proteomics data have been deposited in the ProteomeXchange Consortium via the PRIDE partner repository with the dataset identifier PXD033830 and 10.6019/PXD033830.

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
