# Peer review of "Differentially Represented Proteins in Response to Infection with Mycobacterium tuberculosis Identified by Quantitative Serum Proteomics in Asian Elephants"

_pathogens, 2022, doi:10.3390/pathogens11091010_

Round 1
Reviewer 1 Report
Villar and colleagues determined the proteins expressed in response to tuberculosis in elephants
The authors aimed to discover proteins expressed, which could be used as diagnostic tools. Blood samples were collected from 4 elephants with and without TB and proteomics were performed. The authors identified 26 proteins different between the two groups, from what some were Ig-like proteins and others (16 proteins) were not.
Comments and suggestions:
1. The authors plan to use differential expressed protein for diagnostic of TB. How about other differential diagnostics? They may induce similar proteins, which make difficult to discern.
2. How many times was the proteomics run? Just once or more than once? The authors need to be specific.
3. The part dealing with “candidate protective antigens” is most speculations and should be treat as such. Until today, we do not understand protection against TB, therefore it can be kept in mind when addressing the issue.
4. Animals are exposed to environmental mycobacteria constantly, and it should not be a surprised observation that cross-reactive antigens among the mycobacteria would induce antibody productions.
5. Also, the authors should be careful in state that a protein is part of a protective mechanism when isolated from elephants and compared to blood proteins from animals without active infection. It may be or not.
6. Candidate diagnostic antigens need to be matched against environmental mycobacteria before any conclusions.
Author Response
Reviewer 1
Villar and colleagues determined the proteins expressed in response to tuberculosis in elephants
The authors aimed to discover proteins expressed, which could be used as diagnostic tools. Blood samples were collected from 4 elephants with and without TB and proteomics were performed. The authors identified 26 proteins different between the two groups, from what some were Ig-like proteins and others (16 proteins) were not.
Comments and suggestions:
1.The authors plan to use differential expressed protein for diagnostic of TB. How about other differential diagnostics? They may induce similar proteins, which make difficult to discern.
Response: We understand reviewer comment, and this is the reason we propose the differentially represented antigens and derived chimeric antigen that require additional validation. In response to reviewer comment we added the following statement to Discussion, “However, infection with other pathogens may also affect serum levels of these proteins and thus the proposed candidate diagnostic antigen requires additional experiments for validation in TB versus other common infections in elephants”
- How many times was the proteomics run? Just once or more than once? The authors need to be specific.
Response: As described in M&M, section 2.3, the MS analysis was conducted on 2 technical replicates from each of the 4 biological replicates for TB- and TB+ groups, “For SWATH quantitative analysis, 8 independent samples (2 technical replicates from each of the 4 biological replicates for TB- and TB+ groups) (6 μg each) were subjected to the cyclic data independent acquisition (DIA) of mass spectra using the SWATH variable windows calculator (V 1.0, AB SCIEX) and the SWATH acquisition method editor (AB SCIEX), like established methods [28]”
- The part dealing with “candidate protective antigens” is most speculations and should be treat as such. Until today, we do not understand protection against TB, therefore it can be kept in mind when addressing the issue.
Response: We agree with reviewer comment, and to highlight these limitations the following statement was added to this section, “Based on the challenge that represents a better understanding of the protective mechanisms against TB [66], the proposed candidate protective chimeric antigen is only speculative and requires validation in vaccination studies”
- Animals are exposed to environmental mycobacteria constantly, and it should not be a surprised observation that cross-reactive antigens among the mycobacteria would induce antibody productions.
Response: We agree with reviewer comment and to address it the following statement was added to Discussion, “These animals are exposed to MBTC and environmental mycobacteria such as Mycobacterium simiae and M. avium complex identified here and therefore antibodies induced by bacterial cross-reactive antigens may be present [37]”
- Also, the authors should be careful in state that a protein is part of a protective mechanism when isolated from elephants and compared to blood proteins from animals without active infection. It may be or not.
Response: Again, we agree with reviewer comment. Accordingly, we revised the last paragraph of section 3.3 as “However, the non-Ig proteins identified as overrepresented in TB- elephants and thus underrepresented in TB+ animals were associated with possible protective mechanisms such as defense against infection and reduce risk of inflammation (Fig. 4; Additional files 3 and 4)”, and in Discussion as “Possible protective mechanisms associated with non-Ig proteins overrepresented in TB- elephants included…”
- Candidate diagnostic antigens need to be matched against environmental mycobacteria before any conclusions.
Response: To address reviewer comment, additional analyses were included in section “Candidate diagnostic antigens” and Figure 5, “The proposed candidate diagnostic chimeric antigen protein BLAST sequence alignment against mycobacteria nr database showed 31.6 to 78.9% sequence identity to environmental mycobacteria (Fig. 5). Mycobacteria with highest sequence identity in the M. avium complex also contain pathogenic bacteria found in elephants [34]. These results support the possibility of using this chimeric antigen combined with other tests for the diagnostic of TB in elephants”
Reviewer 2 Report
This paper on elephant tuberculosis uses serum proteomics approach to characterize differentially represented proteins in response to tuberculosis in Asian elephants. The paper has revealed novel finding on antibody response to M. tb infection and biomarkers associated with TB in Asian elephants from Nepal. However, I have following comment:
In line 203, and in Fig 1A, you have mentioned 4 elephants are TB+. In Line 106, you have mentioned 2 elephants were confirmed M. tb positive by cell culture. Are these two elephants culture positive on trunk wash samples? Culture is the gold standard for TB diagnosis in elephants. Elephants reactive on DPP VetTB have antibodies against TB bacteria but they cannot be confirmed TB positive. I would prefer to classify DPP reactive elephants as TB suspected. So, could you please clarify more on case definition on TB + and TB- elephants?
Author Response
Response: As disclosed in the paper, only 2 samples were cultured. The serological test is specific for detection of M. tuberculosis or M. bovis reactive antibodies by using two separate recombinant proteins. All information is disclosed in the Chembio Diagnostic Systems Inc. site. Also a sentence in the Introduction was revised as “The main causative agent for TB in elephants is M. tuberculosis [7] with rare cases of other MBTC organisms such as M. bovis [8]”
Reviewer 3 Report
This is a well-written and interesting paper, which can serve as a guide for additional research and future development of diagnostic, prognostic, and protective tools to monitor and control tuberculosis in Asian elephants.
Minor edits are required throughout – for example:
- Line 71: “in on” should be “on”
- Lines 28 & 82: add an “s” to “advance:
- Line 98: change “…trained veterinarian doctors from the ear…” to “…trained veterinarians from ear veins…”
- Line 358: sentence ends with “may” – delete that word
Author Response
Reviewer 2
This is a well-written and interesting paper, which can serve as a guide for additional research and future development of diagnostic, prognostic, and protective tools to monitor and control tuberculosis in Asian elephants.
Minor edits are required throughout – for example:
Line 71: “in on” should be “on”
Lines 28 & 82: add an “s” to “advance:
Line 98: change “…trained veterinarian doctors from the ear…” to “…trained veterinarians from ear veins…”
Line 358: sentence ends with “may” – delete that word
Response: Thank you for your comments. All edits were corrected, and manuscript revised for additional edits.
Reviewer 4 Report
An interesting study, with potential to influence TB diagnostics. Please find detailed comments in the attached file.
Main comments:
1. The only member of the MTBC identified was Mycobacterium tuberculosis, and only from 2 of the 4 sero-positive elephants. This paper should specify ‘M. tuberculosis infection’, instead of generalising and using the term ‘tuberculosis’
2. The introduction can be used to highlight that M. tuberculosis is most commonly found, and other MTBC only occasionally (mention bTB, and other case reports on rare MTBC organisms, elephants only)
3. After this, continue with the paper on M. tuberculosis. In the discussion, the other MTBC can be discussed again, and how these proteins would identify them (or distinguish between them, which would be the most interesting).
4. Were all 4 sero-positive sera cultured, and only 2 positive, or were only 2 of the 4 cultured? If all 4 were cultured and only 2 positive, please discuss. Include a comparison between the 2 sero-positive + culture postivies and sero-positives + culture negatives if so. If only 2 were cultured, explain why you are assuming the other 2 were M.tuberculosis positive as well. Could it have been another MTBC? Does your serological test differentiate between them?
Structure and images:
1. The authors are requested to ensure only results are published in the results section. The materials and methods should be in their own section and not repeated in results. Any interpretation of the results (and ideas on what the results could be used for) should be in the discussion.
2. The illustrations are superfluous. The results (data) could be displayed e.g. in a table and perhaps for clarity illustrated with graphs. Drawings of elephants do not belong in this manuscript.

Author Response
Reviewer 3
An interesting study, with potential to influence TB diagnostics. Please find detailed comments in the attached file.
Response: Detailed comments in the attached file were considered in the revised manuscript.
Main comments:
1.The only member of the MTBC identified was Mycobacterium tuberculosis, and only from 2 of the 4 sero-positive elephants. This paper should specify ‘M. tuberculosis infection’, instead of generalising and using the term ‘tuberculosis’
Response: Done as suggested.
- The introduction can be used to highlight that M. tuberculosisis most commonly found, and other MTBC only occasionally (mention bTB, and other case reports on rare MTBC organisms, elephants only)
Response: In response to reviewer comment, a sentence in the Introduction was revised as “The main causative agent for TB in elephants is M. tuberculosis [7] with rare cases of other MBTC organisms such as M. bovis [8]”
- After this, continue with the paper on M. tuberculosis. In the discussion, the other MTBC can be discussed again, and how these proteins would identify them (or distinguish between them, which would be the most interesting).
Response: To address this comment and in agreement with Reviewer 1, we revised with new analyses section “Candidate diagnostic antigens” and Figure 5, “The proposed candidate diagnostic chimeric antigen protein BLAST sequence alignment against mycobacteria nr database showed 31.6 to 78.9% sequence identity to environmental mycobacteria (Fig. 5). Mycobacteria with highest sequence identity in the M. avium complex also contain pathogenic bacteria found in elephants [34]. These results support the possibility of using this chimeric antigen combined with other tests for the diagnostic of TB in elephants”, and in Discussion, “These animals are exposed to MBTC and environmental mycobacteria such as Mycobacterium simiae and M. avium complex identified here and therefore antibodies induced by bacterial cross-reactive antigens may be present [37]”
- Were all 4 sero-positive sera cultured, and only 2 positive, or were only 2 of the 4 cultured? If all 4 were cultured and only 2 positive, please discuss. Include a comparison between the 2 sero-positive + culture postivies and sero-positives + culture negatives if so. If only 2 were cultured, explain why you are assuming the other 2 were M.tuberculosispositive as well. Could it have been another MTBC? Does your serological test differentiate between them?
Response: As disclosed in the paper, only 2 samples were cultured. The serological test is specific for detection of M. tuberculosis or M. bovis reactive antibodies by using two separate recombinant proteins. All information is disclosed in the Chembio Diagnostic Systems Inc. site.
Structure and images:
- The authors are requested to ensure only results are published in the results section. The materials and methods should be in their own section and not repeated in results. Any interpretation of the results (and ideas on what the results could be used for) should be in the discussion.
Response: Thanks for reviewer comment. The paper is structured in this way. Only Fig. 1A with a representation of the methodological approach was included in Results in combination with results of the study.
- The illustrations are superfluous. The results (data) could be displayed e.g. in a table and perhaps for clarity illustrated with graphs. Drawings of elephants do not belong in this manuscript.
Response: We don’t agree with this reviewer comment. First, as indicated in the paper, full data is disclosed in Supplementary Materials, including Data S1. Serum proteomics analysis, and also available in the ProteomeXchange Consortium via the PRIDE partner repository with the dataset identifier PXD033830 and 10.6019/PXD033830. The figures as other reviewers agree with are included to facilitate disclosure of complex datasets. We consider that elephant images (not drawings) are courtesy of the authors and appropriate for a paper addressing the study of TB in this species.
Round 2
Reviewer 1 Report
I am satisfied with the authors' answers.
Author Response
Thanks!
Reviewer 4 Report
Dear Authors
It seems that you did not get a chance to view my detailed comments in the first version of your manuscript. Please find them again in attachment.

Author Response
Response: We appreciate reviewer comments, which were addressed as disclosed in the revised manuscript and in the reviewer´s pdf attached.
